# The Role of PIK3R1 in Metabolic Function and Insulin Sensitivity

**DOI:** 10.3390/ijms241612665

**Published:** 2023-08-11

**Authors:** Ariel Tsay, Jen-Chywan Wang

**Affiliations:** 1Metabolic Biology Graduate Program, University of California Berkeley, Berkeley, CA 94720, USA; atsay12@berkeley.edu; 2Department of Nutritional Sciences & Toxicology, University of California Berkeley, Berkeley, CA 94720, USA; 3Endocrinology Graduate Program, University of California Berkeley, Berkeley, CA 94720, USA

**Keywords:** phosphoinositide 3-kinases (PI3K), PIK3R1, p85α, insulin signaling, insulin resistance, type 2 diabetes, metabolic disorders

## Abstract

PIK3R1 (also known as p85α) is a regulatory subunit of phosphoinositide 3-kinases (PI3Ks). PI3K, a heterodimer of a regulatory subunit and a catalytic subunit, phosphorylates phosphatidylinositol into secondary signaling molecules involved in regulating metabolic homeostasis. PI3K converts phosphatidylinositol 4,5-bisphosphate (PIP_2_) to phosphatidylinositol 3,4,5-triphosphate (PIP_3_), which recruits protein kinase AKT to the inner leaflet of the cell membrane to be activated and to participate in various metabolic functions. PIK3R1 stabilizes and inhibits p110 catalytic activity and serves as an adaptor to interact with insulin receptor substrate (IRS) proteins and growth factor receptors. Thus, mutations in PIK3R1 or altered expression of PIK3R1 could modulate the activity of PI3K and result in significant metabolic outcomes. Interestingly, recent studies also found PI3K-independent functions of PIK3R1. Overall, in this article, we will provide an updated review of the metabolic functions of PIK3R1 that includes studies of PIK3R1 in various metabolic tissues using animal models, the mechanisms modulating PIK3R1 activity, and studies on the mutations of human *PIK3R1* gene.

## 1. Introduction

Phosphoinositide 3-kinases (PI3Ks) are signaling molecules that play an imperative role in the regulation of metabolic homeostasis and insulin sensitivity [1]. PI3Ks act downstream of insulin and growth factor receptors to phosphorylate the 3′ hydroxyl group on the inositol head group of a family of acidic phospholipids called phosphatidylinositol (PtdIns) to generate phosphoinositides [2,3,4]. PI3Ks create four different phosphorylated PtdIns: PtdIns3P, PtdIns(3,4)P_2_, PtdIns(3,5)P_2_ and PtdIns(3,4,5)P_3_ (PIP_3_) [4]. These phosphorylated PtdIns act as secondary messengers to modulate various physiological processes. PI3Ks are further classified into three different classes based on the different protein domains and regulatory subunits they contain [5]. Among these three classes, class I PI3Ks play important roles in the regulation of metabolic homeostasis. Class I PI3Ks are categorized as heterodimers consisting of a catalytic p110 subunit and a regulatory subunit [5]. There are four catalytic subunits in human and mouse genomes: p110α, p110β, p110γ and p110δ. Detailed knowledge of the catalytic p110 subunits has been reported in multiple reviews [4,5,6,7,8]. In contrast to the catalytic subunit, the regulatory subunits of PI3Ks are involved in stabilizing the PI3K enzyme as well as regulating its activity [7]. Class IA PI3Ks contain five different p85 or p85-like regulatory subunits. p85α and its splice variants, p55α and p50α, are encoded by the gene *PIK3R1*, while p85β is encoded by *PIK3R2*, and p55γ is encoded by *PIK3R3*. Besides *PIK3R1-3*, there is also *PIK3R5*, which encodes for the regulatory subunit p101, and *PIK3R6*, which encodes for the regulatory subunit p84/87 [7]. p101 and p84/87 are associated with the p110γ catalytic subunit. Together, p101, p84/87 and p110γ make up class IB PI3Ks [7]. Among these regulatory subunits, PIK3R1 has been studied extensively. In this review, we will discuss the current understanding of the role of PIK3R1 in the regulation of metabolic homeostasis. 

PIK3R1, located in chromosome 5 in humans and in chromosome 13 in mice, contains two Src homology 2 (SH2) domains (an N-terminal nSH2 and a C-terminal cSH2 domain). Between these two SH2 domains is a region called the inter-SH2 (iSH2) domain (Figure 1). The nSH2 and cSH2 domains recognize pYXXM motifs on receptor tyrosine kinases or adaptor proteins, such as IRS-1 [9,10]. It has been shown that the iSH2 domain binds to p110α but is not able to affect p110α activity by itself. Rather, the presence of nSH2 is mandatory for PIK3R1 to inhibit p110α and sequester it in the cytosol in the basal state [4,11]. PIK3R1 releases its inhibitory effect on the p110 catalytic subunit when the SH2 domains, especially the nSH2 domain, is bound to phosphorylated receptor tyrosine kinases or adaptor proteins after cellular stimuli [12]. PIK3R1 contains two other domains located at the N terminus of the SH2 domains. These include a SH3 domain as well as a Bcr Homology (BH) domain. The SH3 domain interacts with components of the cytoskeleton, whereas the BH domain interacts with small GTP binding proteins [13,14]. p55α and p50α lack the SH3 and BH domain. Instead, p55α contains a 34-amino-acid region followed by a conserved proline-rich motif, while p50α contains a 6-amino-acid region followed by a conserved proline-rich motif [7,15]. p85α is abundantly expressed in several tissues like brain, liver, muscle, fat and kidney. While p50α and p55α are both expressed in fat, only p50α is expressed in liver and muscle as well [15,16]. 

PIK3R1 has tumor suppressor potential, and its mutations have been associated with several cancers. Specifically, reduced levels of PIK3R1 levels due to mutations, commonly in the iSH2 or nSH2 domain, or deletions result in constant activation of the PI3K pathway [17]. This leads to the activation of downstream AKT signaling which can induce carcinogenesis [18]. This dysregulated mechanism of PI3K has been shown to be the cause for numerous cancers [17,19]. Overall, several articles have provided extensive reviews on this topic [17,18,20]. Notably, numerous PI3K inhibitors have been designed with the purpose to be used as cancer treatment [21,22]. However, these inhibitors either target the p110 isoforms or downstream of PI3K signaling because p85α is not known to be druggable [18,21]. These PI3K inhibitors result in several adverse effects, with hyperglycemia being a common outcome [21,22]. 

PIK3R1 undergoes various post-translational modifications such as phosphorylation, SUMOylation by small ubiquitin-like modifier (SUMO) proteins, and ubiquitination. Several phosphorylation sites in PIK3R1 have been identified. Phosphorylation at Ser361 and Ser652 in the SH2 domains of mouse Pik3r1 protein prevent its binding to tyrosine-phosphorylated peptides [23]. While Ser652 has been shown to be phosphorylated by protein kinase D (PKD), Ser361 is suggested to be phosphorylated by a protein kinase C family member [23]. Furthermore, human PIK3R1 protein can be phosphorylated at Ser608 by the kinase activity of the p110 catalytic subunit, which results in decreased PI3K activity [24]. This inhibitory action, however, can be reversed by protein phosphatase 2A [25]. In addition, the phosphorylation at Tyr458 of PIK3R1 by Src family kinases in head and neck cancer cells leads to increased PI3K activity [26,27]. Another study showed that Src family kinases LCK and ABL phosphorylate the cSH2 domain of PIK3R1 at Tyr688 in vitro, which increases PI3K activity due to the release of the inhibitory effect of p85α on p110 [12,28]. On the other hand, SHP-1, a non-receptor tyrosine phosphatase, dephosphorylates PIK3R1 at Tyr688 and leads to a decrease in PI3K activity [29]. PKA has been shown to phosphorylate PIK3R1 at Ser83, which is necessary for cAMP-PKA-induced G1 arrest and survival of 3T3 fibroblasts [30]. This phosphorylation also plays a role in the control of MCF-7 human breast cancer cell proliferation and retinoic acid-induced inhibition of proliferation and motility [31]. It has been suggested that PIK3R1 contains sequences similar to that of SUMO consensus sites [32]. One study confirms that PIK3R1 is conjugated to SUMO1 and SUMO2 and that lysine residues within the iSH2 domain of PIK3R1 likely contain the SUMOylation sites [32]. There is also an inverse correlation between PIK3R1 SUMOylation and PIK3R1 phosphorylation at Tyr458 [32]. PIK3R1 has also been shown to be ubiquitinated. This is discussed in the section that introduces the PIK3R1-interacting proteins below. Notably, how these post-translational modifications contribute to the role of PIK3R1 in metabolic regulation is mostly unclear.

**Figure 1 ijms-24-12665-f001:**
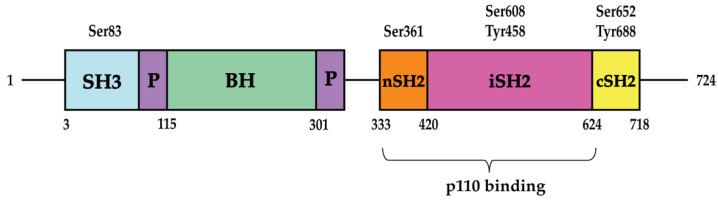
The structure of p85α. Several phosphorylation sites have been identified in the different domains of p85α. In the basal state, the nSH2 domain interacts with p110, resulting in p110 inhibition.

## 2. Insulin Signaling 

PIK3R1’s role in the regulation of insulin signaling plays an important function in metabolic homeostasis (Figure 2) [33]. The insulin receptor (IR) is an allosteric enzyme and a receptor tyrosine kinase (RTK) [33]. When insulin binds to IR, the receptor dimerizes and becomes activated [33]. It autophosphorylates multiple tyrosine residues that in turn act as docking sites for proteins that contain SH2 or PTB domains, such as IRS proteins [33,34,35]. After recruiting IRS proteins, IR phosphorylates them at multiple tyrosine residues [1]. These phosphorylated IRS proteins in turn recruit several signaling proteins through interaction with the SH2 domain of these downstream signaling molecules, such as PIK3R1 [1]. After binding to IRS proteins, PIK3R1 releases its inhibitory effect on the p110 catalytic subunit, allowing the p110 subunit to move closer to the plasma membrane to convert PIP_2_ to PIP_3_ [4,12]. High levels of PIP_3_ are highly regulated by phosphatase and tensin homolog (PTEN), which converts PIP_3_ back to PIP_2_ [36]. PIP_3_ induces the recruitment of other proteins with a plekstrin homology (PH) domain, such as PDK1 and AKT. When PDK1 and AKT are recruited to the plasma membrane, this leads to PDK1 phosphorylating AKT on the Thr308 residue [37]. This phosphorylation event activates AKT, resulting in activation of downstream AKT-mediated pathways. In myotubes and adipocytes, AKT phosphorylates AS160, which plays a key role in the translocation of the glucose transporter GLUT4 from the cytoplasmic GLUT4 storage vesicles onto the surface of the cell membrane [38]. This increases glucose transport into the cells. In hepatocytes, activated AKT phosphorylates and inhibits GSK3β, which in turn activates glycogen synthase [1]. Activated AKT is also important for suppressing the transcription of gluconeogenic genes, such as *phosphoenolpyruvate carboxykinase* (*PCK1*) and the *catalytic subunit 1 of glucose 6 phosphatase* (*G6PC1*). This is achieved by phosphorylating the FOXO1 transcription factor and excluding it from the nucleus to the cytosol. While PIK3R1 plays an important role as a regulatory subunit of PI3K in these insulin-regulated metabolic functions, it is important to note that excess amounts of monomeric PIK3R1 can compete with PI3K heterodimer (consisting of PIK3R1 and p110) for binding to IRS proteins [9]. As a result, this competition leads to the inhibition of insulin signaling. On the other hand, reducing PIK3R1 levels has been suggested to increase PI3K heterodimer recruitment to RTKs and increase PI3K signaling to induce AKT activity [39]. PIK3R1 also protects PTEN from degradation. This is discussed in the section below.

## 3. Metabolic Functions of PIK3R1 In Vivo 

Animal models have been created to study the metabolic function of PIK3R1 (Table 1). Homozygous deletion of all *Pik3r1* isoforms (p85α, p55α, and p50α) in mice with a mixed genetic background (129SvEv x C57Bl/6) or inbred 129SvEv background leads to perinatal lethality. These mice exhibit extensive hepatocyte necrosis and chylous ascites [40]. Other phenotypes include enlarged skeletal muscle fibers, brown fat necrosis and calcification of cardiac tissue [40]. In liver and muscle, there is a significant decrease in the expression and the activity of the p110 catalytic subunit, which leads to an 80–90% decrease in total class IA PI3K activity [40]. However, these homozygous mice are still hypoglycemic and have lower insulin levels in the fasted and fed state [40]. During an intraperitoneal glucose tolerance test (GTT), these mice exhibit faster glucose clearance while still maintaining lower insulin levels [40]. Interestingly, mice lacking only p85α (*Pik3r1*^−/−^) but retaining the p50α and p55α isoforms are hypoglycemic and have lower insulin levels in fed and fasted states compared to wild-type mice [41]. These *Pik3r1*^−/−^ mice exhibit an isoform switch from p85α to p50α in skeletal muscle and adipocytes [41]. In adipocytes of *Pik3r1*^−/−^ mice, the binding of p50α to p110α contributes to an increase in PIP_3_ levels and facilitates the translocation of GLUT4 to the plasma membrane [41]. Consequently, this leads to increased glucose transport and hypoglycemia. To compensate for the increase in glucose transport into adipocytes, *Pik3r1*^−/−^ mice have increased serum leptin production to prevent adiposity [42]. This suggests that *Pik3r1*^−/−^ mice exhibit mild leptin resistance as more leptin is needed to maintain similar weight and fat mass as wild-type mice [42]. When fed on a high-fat diet (HFD), *Pik3r1*^−/−^ mice still show improved glucose tolerance and better insulin sensitivity compared to wild-type mice on an HFD [42]. While both groups have a similar induction of leptin secretion, *Pik3r1*^−/−^ mice have increased WAT mass and body weight compared to wild-type mice on an HFD [42]. Therefore, unlike when on a normal diet, the increase in leptin secretion in *Pik3r1*^−/−^ mice on an HFD is not enough to reduce WAT adiposity [42]. In addition, *Pik3r1*^−/−^ mice on a normal diet also show increased serum adiponectin levels [42]. However, the mechanism of how *Pik3r1* regulates adiponectin levels still needs to be further investigated. All together, these results indicate that p50α and p55α both are involved in glucose and lipid metabolism. Moreover, a decrease in class IA PI3K activity does not result in insulin resistance.

Leptin has also been shown to alter p85α binding to IRS proteins [4,43]. Fao hepatoma cells treated with leptin and then stimulated with insulin demonstrated an increase in p85α binding to IRS-1 and a significant decrease in p85α binding to IRS-2 [43]. This suggests that leptin may play a role in distinguishing what IRS protein p85α will bind to. Furthermore, while leptin treatment in Fao hepatoma cells induces AKT and GSK3 phosphorylation, it results in no additive effect to insulin-mediated phosphorylation of AKT and GSK3 [43]. 

Mice with the deletion of the two isoforms of *Pik3r1*, p55α and p50α, have been generated [16]. These mice are viable and have normal blood glucose levels but have lower fasting insulin levels [16]. Insulin tolerance test (ITT) shows that they have enhanced insulin sensitivity, which is accompanied by an increase in insulin-stimulated glucose transport in isolated extensor digitorum longus muscle and adipocytes [16]. In muscle, insulin-stimulated IRS-1 and phosphotyrosine-associated PI3K is reduced, but IRS-2-associated PI3K and AKT activation is enhanced [16]. The adipocytes of these mice are smaller in size due to having lower lipid content, as they exhibit decreased basal lipogenesis [16]. However, insulin-stimulated PI3K and AKT are not affected in adipocytes [16]. The increased glucose uptake in adipocytes is likely due to other mechanisms, such as increased lactate synthesis in both the basal and insulin-stimulated conditions [16]. These studies indicate that p50α and p55α both are involved in insulin signaling and thus are imperative in maintaining glucose and lipid homeostasis. Several *Pik3r1* tissue-specific deletion animal models have been generated to further identify the role of *Pik3r1* in specific tissues (Figure 3). 

### 3.1. Liver Pik3r1 

*Pik3r1^flox/flox^* mice were crossed with mice expressing Cre recombinase under control of the albumin promoter to create liver-specific *Pik3r1* knockout mice (L-*Pik3r1*^−/−^). These mice are viable and are shown to have improved hepatic and whole-body insulin sensitivity [44]. L-*Pik3r1*^−/−^ mice have an 80–90% decrease in p85α and a complete loss of p50α (the only spliced isoform present in the liver) [44]. There is a 50% reduction in IRS-1-associated PI3K activity as well as a 70% decrease in the expression of p110α. However, AKT activity was upregulated in L-*Pik3r1*^−/−^ mice. This is due to increased PIP_3_ accumulation in the liver of L-*Pik3r1*^−/−^ mice compared to wild-type mice [44]. Notably, L-*Pik3r1*^−/−^ mice have decreased PTEN activity in the liver despite there being no changes in PTEN expression [44]. It has been shown that p85α can bind directly to and enhance PTEN lipid phosphatase activity [45]. Moreover, p110α-free p85α homodimerizes through two intermolecular interactions (SH3:proline-rich region and BH:BH regions) to selectively bind to unphosphorylated and activated PTEN [46]. This homodimeric p85α inhibits the PI3K pathway by preventing E3 ligase WWP2-mediated proteasomal degradation of PTEN [46]. Additionally, L-*Pik3r1*^−/−^ mice on an HFD are protected against c-Jun N-terminal kinase (JNK)-induced insulin resistance [47]. In this case, the role of p85α for JNK activation is distinct from its function as a component of PI3K. It occurs specifically in response to insulin and endoplasmic reticulum (ER) stress [47]. The activation of JNK by p85α is mediated through CDC42 and MKK4, which requires both the presence of the N terminus and functional SH2 domains within the C terminus of p85α [47]. Therefore, these studies suggest that *Pik3r1* regulates hepatic insulin sensitivity. Furthermore, *Pik3r1* itself is involved in regulating the PI3K pathway through maintaining or enhancing PTEN activity. 

### 3.2. Skeletal Muscle Pik3r1 

*Pik3r1^flox/flox^* mice were crossed with mice expressing Cre recombinase under control of the muscle creatine kinase gene promoter to generate striated muscle-specific *Pik3r1* knockout (M-*Pik3r1*^−/−^). Notably, all three isoforms of Pik3r1 are deleted in both skeletal muscle and cardiac muscle in these mice. M-*Pik3r1*^−/−^ mice have decreased insulin-stimulated IRS-1- and IRS-2-associated PI3K activity in gastrocnemius muscle [48]. AKT activation, interestingly, was not decreased in these mice. When M-*Pik3r1*^−/−^ mice were crossed with *Pik3r2*^−/−^ mice (M-*Pik3r1*^−/−^/*Pik3r2*^−/−^) to also delete *Pik3r2* in the muscle, these mice have impaired insulin-stimulated AKT1 and AKT2 activity in the gastrocnemius muscle [48]. This suggests that deletion of *Pik3r1* alone in striated muscle is enough to disrupt PI3K signaling, but it does not severely affect AKT activation. These M-*Pik3r1^−/−^/Pik3r2*^−/−^ mice are glucose intolerant due to muscle insulin resistance [48]. Despite having impaired glucose disposal, reduced insulin release during GTT, and decreased insulin sensitivity, M-*Pik3r1^−/−^/Pik3r2*^−/−^ mice have normal fasting and fed blood glucose and insulin levels [48]. Moreover, ITT shows that HFD feeding in M-*Pik3r1*^−/−^/*Pik3r2*^−/−^ mice does not exacerbate insulin resistance compared to wild-type mice [48]. This suggests that insulin resistance in muscle is enough to cause glucose intolerance, but not hyperglycemia [48]. This may be due to the presence of other insulin sensitive tissues [48]. Furthermore, M-*Pik3r1^−/−^/Pik3r2*^−/−^ mice also have reduced muscle weight and myocyte size [48]. Thus, together, muscle *Pik3r1* and *Pik3r2* are important for maintaining muscle insulin sensitivity and glucose tolerance but seem to have a minor role in maintaining whole-body glucose homeostasis due to other insulin sensitive tissues like liver and adipose in the body. 

### 3.3. Adipose Pik3r1

It has been shown that all *Pik3r1* isoforms are highly induced in white adipose tissue (WAT) of HFD-fed obese mice [49]. Whole-body heterozygous deletion of the *Pik3r1* regulatory subunits (*Pik3r1*^+/−^), but not knockout of *Pik3r2*, preserves whole-body WAT and skeletal muscle insulin sensitivity, despite severe obesity [49]. Moreover, WAT macrophage accumulation, proinflammatory gene expression, and ex vivo chemokine secretion in obese *Pik3r1*^+/−^ mice are markedly reduced despite endoplasmic reticulum (ER) stress, hypoxia, adipocyte hypertrophy, and JNK activation [49]. Furthermore, bone marrow transplant studies reveal that these improvements in obese *Pik3r1*^+/−^ mice are independent of reduced *Pik3r1* expression in the hematopoietic compartment [49]. 

*Pik3r1^flox/flox^* mice were crossed with mice expressing Cre recombinase under the control of the *Ucp1* gene promoter to generate brown adipocyte-specific *Pik3r1* knockout mice (BAT-*Pik3r1*^−/−^) [50]. These BAT-*Pik3r1*^−/−^ mice have improved thermogenic functions and reduced HFD-induced adiposity and body weight, insulin resistance and hepatic steatosis [50]. These phenotypes are suggested to be due to lower activation of JNK in BAT as well as enhanced insulin receptor isoform B expression and association with IRS-1 in BAT [50]. They also have lower levels of proinflammatory cytokines in BAT and inguinal WAT (iWAT) [50]. Moreover, the browning of iWAT of these mice is also increased [50]. Thus, *Pik3r1* expression in adipocytes plays a critical role in mediating WAT and BAT insulin sensitivity, and reduced PI3K activity is a key step in the initiation and propagation of the inflammatory response in WAT of obese animals. 

### 3.4. β Cell Pik3r1

*Pik3r1^flox/flox^* mice were crossed with mice expressing Cre recombinase under control of the rat insulin promoter (RIP-Cre) to generate β cell-specific *Pik3r1* knockout mice (β-*Pik3r1*^−/−^) [51]. β-*Pik3r1*^−/−^ mice are more glucose intolerant and have impaired insulin secretion [51]. They do not show any difference in insulin sensitivity during ITT compared to wild-type mice [51]. Furthermore, glucose-stimulated insulin secretion tests along with pancreas perfusion tests show that glucose-stimulated insulin secretion (GSIS) is reduced in β-*Pik3r1*^−/−^ mice [51]. Thus, the deletion of p85α in β cells results in glucose intolerance due to impaired insulin secretion. These mice also have blunted PI3K/AKT signaling as well as a decrease in the ratio of islet area to total pancreas at 32 weeks of age [51]. To ensure complete deletion of any downstream PI3K signaling, mice with β cell-specific deletion of *Pik3r1* and *Pik3r2* were generated. These mice show exacerbated glucose intolerance and defect in GSIS when compared to β-*Pik3r1*^−/−^ mice [51]. The impaired GSIS in these mice is likely due to the loss of synchronicity in β cell insulin secretion and diminished exocytosis of insulin caused by reduction in the expression of SNARE complex genes [51]. Overall, these results demonstrate that Pik3r1 plays a role in maintaining insulin secretion in β cells. 

Akita mice develop T1DM due to a mutation in the insulin 2 (*Ins2*) gene. This mutation causes the abnormal folding of insulin protein that leads to ER stress and thus chronically activates the unfolded protein response (UPR) [52]. While the function of the UPR is to alleviate ER stress, chronic UPR activation can lead to the apoptosis and dysfunction of β cells [31]. Akita^+/−^ mice were crossed with β-*Pik3r1*^−/−^ to generate Akita^+/−^/β-*Pik3r1*^−/−^ mice [52]. These mice do not develop hyperglycemia or reduced plasma insulin levels, which are observed in Akita^+/−^ mice in either fed or fasted states [52]. This suggests that *Pik3r1* in β cells of Akita^+/−^ mice contribute to reduced β cell function and mass under the chronic ER stress conditions [52]. While β cells of Akita^+/−^ mice show signs of ER stress, such as a distended ER, as well as reduced insulin secretory granules, Akita^+/−^/β-*Pik3r1*^−/−^ mice have a normal ER structure and abundant insulin secretory granules [52]. Islets of Akita^+/−^/β-*Pik3r1*^−/−^ mice also have reduced apoptotic rates by the UPR-activated NLRP3 inflammasome pathway, ER stress and oxidative stress compared to Akita^+/−^ mice [52]. Notably, p85α has been shown to be involved in the UPR by interacting with X-box binding protein-1 (XBP-1s) [53]. In murine MIN6 pancreatic β cells, the overexpression of p85α stabilizes XBP-1 protein levels and potentiates XBP-1-dependent apoptosis [52]. These results suggest that under chronic ER stress conditions, Pik3r1 in β cells is involved in inducing the UPR-mediated apoptosis of β cells, leading to β cell dysfunction.

**Table 1 ijms-24-12665-t001:** Summary of different *Pik3r1* mouse models generated and their characteristics.

Mouse Model	Tissue with *Pik3r1* Knockout	Characteristics	References
Deletion of all *Pik3r1* isoforms	Whole-body homozygous	Causes perinatal lethality and has hepatocyte necrosis, chylous ascites, enlarged skeletal muscle fibers, brown fat necrosis, and calcification of cardiac tissue [40]. Results in a decrease in class 1A PI3K activity in liver and muscle [40]. Is hypoglycemic and has lower insulin levels in fasted and fed state [40]. More glucose tolerant while having lower insulin levels during GTT [40].	Fruman et al. [40]
*Pik3r1* ^+/−^	Whole-body heterozygous	Maintains whole-body insulin sensitivity in WAT and skeletal muscle under HFD [49]. Reduced WAT macrophage accumulation and proinflammatory gene expression [49].	McCurdy et al. [49]
*Pik3r1*^−/−^ retaining p55α and p50α isoforms	Whole-body homozygous	Hypoglycemic due to increased glucose transport [41]. Has lower insulin levels in fed and fasted state [41]. Shows isoform switch to p50α in muscle and adipocyte [41]. Exhibit leptin resistance on normal diet.	Terauchi et al. [41] Terauchi et al. [42]
Deletion of p55α and p50α	Whole-body homozygous	Lower fasting insulin levels, enhanced insulin sensitivity and increased glucose-stimulated glucose transport [16]. Reduced insulin-stimulated IRS-1 phosphotyrosine-associated PI3K but increased IRS-2-associated PI3K and AKT activation [16]. Adipocytes are more insulin sensitive and have lower lipid content [16].	Chen et al. [16]
L-*Pik3r1*^−/−^	Liver	Improved hepatic and whole-body insulin sensitivity [44]. On an HFD, is protected against JNK-induced insulin resistance [47].	Taniguchi et al. [44] Taniguchi et al. [47]
M-*Pik3r1*^−/−^	Skeletal and Cardiac Muscle	Reduced insulin-stimulated IRS-1- and IRS-2-mediated PI3K [48]. Under Dexamethasone (Dex) treatment, have lower Dex-induced impaired AKT activity and lower Dex-induced glucose and insulin intolerance [54]. Attenuated Dex-induced muscle atrophy and Dex-induced inhibition of eIF2α and 4E-BP1 phosphorylation [54].	Luo et al. [48] Chen et al. [54]
M-*Pik3r1*^−/−^/*Pik3r2*^−/−^	Skeletal and Cardiac Muscle	Muscle insulin resistant and has impaired glucose disposal, reduced insulin release and decreased insulin sensitivity [48]. Normal fasting and fed blood glucose and serum insulin levels [48]. On an HFD, does not have exacerbated insulin resistance compared to wild type during ITT [48].	Luo et al. [48]
BAT-*Pik3r1*^−/−^	Brown Adipose Tissue	Improved thermogenic functions, reduced HFD-induced adiposity and body weight, insulin resistance, and hepatic steatosis [50].	Gomez-Hernandez et al. [50]
β-*Pik3r1*^−/−^	β Cell	Glucose intolerant and impaired glucose-stimulated insulin secretion [51].	Kaneko et al. [51]
β-*Pik3r1*^−/−^/*Pik3r2*^−/−^-	β Cell	Exacerbated glucose intolerance and defect in insulin secretion compared to β-*Pik3r1*^−/−^ mice [51]. Loss of synchronicity in β cell insulin secretion and impaired exocytosis of insulin caused by reduced expression of SNARE complex genes [51].	Kaneko et al. [51]
Akita^+/−^/β-*Pik3r1*^−/−^	β Cell deletion in Akita^+/−^ mice	Compared to Akita^+/−^ mice, do not have hyperglycemia or reduced plasma insulin levels [52]. Have normal ER structure and many insulin secretory granules as well as reduced apoptotic rates, ER stress, and oxidative stress [52].	Winnay et al. [52]
*Pik3r1*^+/R649W^ (Arg649Trp mutation)	Whole-body heterozygous	Show phenotypes similar with SHORT syndrome [55]. Are insulin resistant, glucose intolerant, and hyperinsulinemic in fed and fasted states [55]. Have impaired insulin secretion and GLP-1 action [55]. Reduced insulin-stimulated IRS-1 tyrosine phosphorylation and AKT phosphorylation [55]. On an HFD, have lower adiposity but are more hyperglycemic and insulin resistant [55].	Winnay et al. [55]
*Pik3r1*^+/R649W^-*ob*/*ob*	Whole-body heterozygous	Protected from obesity and hepatic steatosis but are hyperglycemic [56].	Solheim et al. [56]
A-*Pik3r1*^−/−^	Adipose	Reduced levels of Dex-induced phospho-Hsl, phospho-Plin1, catalytic and regulatory subunits of PKA [57]. Improved Dex-induced hepatic steatosis and hypertriglyceridemia [57].	Kuo et al. [57]

## 4. GWAS Studies in *PIK3R1*


Multiple mutations in human *PIK3R1* have been associated with SHORT syndrome. SHORT syndrome is a rare autosomal condition consisting of several different features that together form the acronym SHORT. The features include short stature, hyperextensibility of joints, ocular depression, Rieger anomaly and teething delay [58]. The diagnosis, morbidity and mortality of SHORT syndrome are still unknown, and there is no treatment today. Currently, there are around 10 mutations in *PIK3R1* confirmed to be associated with SHORT syndrome that contribute to insulin resistance and/or lipodystrophy [58]. These mutations include missense mutations (converting glutamic acid 489 to lysine, arginine 631 to glutamine, arginine 649 to tryptophan), deletions (deleting isoleucine at amino acid 539), truncation (at tyrosine 657) and frame shift insertions (generating a stop codon at amino acid residue 654 in exon 14, in-frame stop codon 7 residues downstream of amino acid 643, a single nucleotide insertion resulting in a premature truncated protein) [56,59,60]. The missense mutation of arginine 649 to tryptophan (Arg649Trp) is the most common mutation [58]. The Arg649Trp mutation occurs in the cSH2 domain of the *PIK3R1* [61]. The function of the arginine residue is predicted to recognize and bind to tyrosine-phosphorylated substrates [61]. Structural modeling predicts that when arginine is mutated into tryptophan, it prevents PIK3R1 from interacting with its substrates [61]. 

Mice with the Arg649Trp mutation have been generated. Homozygous *Pik3r1*^R649W/R649W^ mice are embryonically lethal [40,55]. The heterozygous male mice *(Pik3r1*^+/R649W^) show several phenotypes that are similar to SHORT syndrome phenotypes (Table 1). They have significantly decreased body weight up to 6 months of age and decreased body length at 12 weeks of age [55]. They also have smaller subcutaneous adipose tissue, while no significant changes are observed in epididymal adipose tissue or brown adipose tissue. The pattern of adipose tissue loss is similar to what is observed in SHORT syndrome patients, who display partial lipodystrophy shown by having reduced subcutaneous fat [55]. The *Pik3r1*^+/R649W^ mice are also insulin resistant, glucose intolerant, and hyperinsulinemic in fed and fasted states while also exhibiting impaired insulin secretion and GLP-1 action on islets in vivo and in vitro [55]. Notably, while most patients with SHORT syndrome are insulin resistant, this symptom is not found in all patients [55,59]. Insulin-stimulated IRS-1 tyrosine phosphorylation is decreased in WAT, liver, and skeletal muscle of *Pik3r1*^+/R649W^ mice. Insulin-stimulated AKT phosphorylation is reduced in the WAT and the liver of these mice [55]. As predicated in structure studies, the R649W mutation impairs p85α binding to phosphorylated motifs on IRS-1, thus reducing the formation of the p85α and IRS-1 complex [55]. 

*Pik3r1*^+/R649W^ mice on an HFD have lower weight gain and adiposity than those of wild-type mice on an HFD [56]. These characteristics are likely caused by significantly lower expression of *fatty acid synthase* (*Fasn*), *hormone sensitive lipase* (*Hsl*) and *ATP citrate lyase* (*Acl*) in the iWAT of *Pik3r1*^+/R649W^ mice [56]. However, even though HFD-fed *Pik3r1*^+/R649W^ mice have lower adiposity, they are more hyperinsulinemic, hyperglycemic and insulin-resistant compared to wild-type mice on an HFD [56]. At 8 weeks of HFD, *Pik3r1*^+/R649W^ mice have reduced phosphorylated IR and phosphorylated AKT in the liver after insulin injection [56]. When *Pik3r1*^+/R649W^ mice were crossed with *ob*/*ob* mice, *Pik3r1*^+/R649W^-*ob*/*ob* mice do not develop obesity and hepatic steatosis (Table 1). However, they are more hyperglycemic compared to *ob*/*ob* mice and retain similar hyperinsulinemic levels exhibited in HFD-fed *Pik3r1*^+/R649W^ mice [56]. Thus, the R649W mutation shows a dissociated function of Pik3r1 on the regulation of lipid and glucose homeostasis. Interestingly, heterozygous *Pik3r1*^+/*−*^ mice are also protected from obesity; however, in contrast to *Pik3r1*^+/R649W^ mice, they remain insulin sensitive [41]. 

Besides often being insulin resistant, SHORT syndrome patients with mutations in *PIK3R1* do not exhibit fatty liver [56,62]. This is different from patients who carry mutations in AKT2, as they develop insulin resistance along with dyslipidemia and fatty liver [63]. Interestingly, HFD-fed *Pik3r1*^+/R649W^ mice develop fatty liver. However, histological evaluation found that hepatic lipid accumulation in these mice is lower compared to the HFD-fed wild-type mice [56]. Under HFD feeding, *Pik3r1*^+/R649W^ mice also have increased mRNA levels of genes involved in β oxidation, such as *carnitine palmitoyl transferase 1* (*Cpt1*), *malonyl-CoA decarboxylase* (*Mcd*) and *acyl-CoA dehydrogenase* (*Acad*), compared to wild-type mice on an HFD [56]. Liver inflammation was also greatly increased in *Pik3r1*^+/R649W^ HFD-fed mice. *Pik3r1*^+/R649W^-*ob*/*ob* mice do not show an increase in liver weight or hepatic triglyceride levels compared to *ob/ob* mice [56]. While the expression of hepatic lipogenic genes, such as *ATP citrate lyase* (*Acl*), *Acetyl-CoA Carboxylase α* (*Acaca*) and *Fasn*, are higher in *ob/ob* mice than that of wild-type mice, this phenotype is not observed in *Pik3r1*^+/R649W^-*ob/ob* mice [56]. There is also no significant difference in liver inflammation for *ob/ob-Pik3r1*^+/R649W^ mice compared to *ob/ob* mice [56]. Thus, the R649W mutation is protective against the development of fatty liver in *ob/ob* mice.

Other mutations in the C-terminal of *PI3KR1* have been reported to result in metabolic and cellular phenotypes. Along with the R649W mutation, a heterozygous Y657X mutation was identified in a patient exhibiting SHORT syndrome and has been identified as the cause for SHORT syndrome [64]. The Y657X mutation, a nonsense mutation, is predicted to remove two-thirds of the cSH2 domain of *PIK3R1* [64]. Fibroblasts obtained from the patient express full-length p85α as well as truncated p85, p55α and p50α [64]. However, the expression of full-length p85α in the patient is lower [64]. Immunoprecipitation from the fibroblasts showed that full-length p85α is still able to interact with p110α. However, the mutant p85α is not detected in the immunoprecipitates [64]. In addition, while phosphotyrosine immunoprecipitation showed an increase in the amount of full-length p85α in phosphotyrosine immunoprecipitates after insulin stimulation in the controls, no such increase in full-length p85 was observed in fibroblasts from the patient [64]. This suggests that binding of mutant p85α to tyrosine phosphorylated IRS-1 is significantly reduced [64]. The overexpression of p85α with the Y657X mutation in 3T3-L1 murine preadipocytes impaired differentiation to adipocytes and reduced AKT phosphorylation by insulin [64]. These results indicate that the Y657X mutation impairs insulin action.

## 5. PIK3R1-Interacting Proteins and PI3K-Independent Function of PIK3R1

Several proteins have been shown to interact with PIK3R1 and modulate its levels (Figure 4). However, the role of many of these interactions in metabolic regulation remain to be evaluated (Table 2). CBL and CBL-B, which are E3 ligases, interact with p85α and lead it to ubiquitination [65,66]. The SH3 domain of p85α mediates the interaction with CBL-B and is required for p85α ubiquitination [66]. The short isoform of ErbB3-binding protein 1 (EBP1), p42, interacts with the cSH2 domain of p85α, leading p85α to the HSP70/CHIP complex for degradation and thus inhibiting PI3K activity [67]. In prostate cancer cells, TGF-β activates the PI3K-AKT pathway to drive cell migration. This process requires the activity of the E3 ubiquitin ligase tumor necrosis factor receptor-associated factor 6 (TRAF6). TRAF6 polyubiquitylates p85α in its iSH2 domain and promotes the formation of the complex between the TGF-β type I receptor (TβRI) and p85α, which leads to the activation of PI3K and AKT [68]. Finally, Kelch repeat and BTB (POZ) domain containing 2 (KBTBD2) is a substrate recognition subunit of the Cullin-3 (CUL3)-based E3 ubiquitin ligase, whose expression is down-regulated in diet-induced obese insulin-resistant mice in a leptin-dependent manner [69]. KBTBD2 interacts with CUL3 and the iSH2 domain of p85α and leads to the degradation of p85α [69]. In the absence of KBTBD2, p85α accumulates to 30-fold greater levels than in wild-type adipocytes, and excessive p110-free p85α blocks the binding of p85α-p110 heterodimers to IRS-1, interrupting insulin signaling [69]. *Kbtbd2*^−/−^ mice exhibit diabetes and hepatic steatosis phenotypes [69]. The homozygous germ line inactivation of *Pik3r1* improves these phenotypes [69].

Some proteins interact with PIK3R1 and modulate its activity. Bromodomain-containing protein 7 (BRD7) regulates PI3K activity through the interaction of p85α [70]. BRD7 directly interacts with the iSH2 domain of p85α, leading to nuclear translocation of p85α [70,71]. HeLa cells overexpressing BRD7 have decreased levels of p85α in the cytosol and therefore have a reduction in PI3K downstream signaling after insulin treatment [70]. This is due to the rapid degradation and denaturation of the p110 catalytic subunit without p85α [70]. On the other hand, when endogenous BRD7 is knocked down with RNAi in HeLa cells, this results in an increase in PI3K signaling. This is due to an increase in the p110 subunit and the localization of more p85α in the cytosol [70]. Interestingly, in the liver, p85α is necessary for BRD7 protein expression since mice that lack p85α have reduced BRD7 protein levels [72]. In pancreatic cancer cells, p85α has been shown to interact with protein kinase PAK4, through the SH3 domain [73]. PAK4 expression is associated with the migration and invasion of pancreatic cancer cells induced by hepatocyte growth factor (HGF) [73]. It is proposed that the interaction of PAK4 with p85α relieves the p110 catalytic subunit of PI3K, which is involved in the migratory response of PAK4. Indeed, PAK4-depleted cells show reduced phosphorylation of AKT [73]. 

In addition to its role in the regulation of PI3K activity, PIK3R1 exerts other actions to modulate metabolic homeostasis. PIK3R1 has been shown to play a role in insulin and ER stress activation of JNK [47]. JNK induces the phosphorylation of IRS-1 at Ser307, which prevents IRS-1 binding to activated IR [74]. L-*Pik3r1*^−/−^ mice have impaired hepatic JNK activation and improved whole-body insulin sensitivity under HFD treatment [47]. PIK3R1 is found to regulate JNK activation by reducing the levels of upstream JNK activators [47]. L-*Pik3r1*^−/−^ hepatocytes have a 65% reduction in activated CDC42 as well as a 75% reduction in MKK4 activation [47,75]. CDC42 is a small GTPase known to activate SEK1/MKK4, which is upstream of JNK [47]. While CDC42 can also be activated by the PI3K pathway, inhibiting PI3K in wild-type cells does not prevent insulin-induced JNK activation, and L-*Pik3r1*^−/−^ cells have a 75% decrease in insulin-induced JNK activation [47]. Thus, instead of PI3K activity, an intact N-terminus and the SH2 domain within the C-terminus of p85α are vital for CDC42 to activate JNK under insulin treatment [47]. Notably, a direct p85α-CDC42 interaction has been reported [76]. Platelet-derived growth factor (PDGF), which is involved in cell migratory response, promotes CDC42 activation as well as its transient interaction with p85α [76]. Furthermore, p85α binding to CDC42 has been found to be involved in cell migration through increasing filopodia formation, N-WASP-mediated decrease in actin stress fibers, and lowering focal adhesion complexes [76]. These functions of p85α require the presence of the BH domain as p50α is unable to bind to CDC42 and does not cause any of the above effects [76]. 

As discussed above, p85α has been shown to interact with the spliced form of XBP-1s and increase the stability of XBP-1s as well as its nuclear translocation [53,77,78]. The interaction between the BH domain of p85α and the spliced form of XBP-1 (XBP-1s) is reduced in the liver of *ob/ob* mice. This reduction leads to a decreased nuclear localization of XBP-1s and an attenuated resolution of the UPR [53,77]. In p85α-deficient cells, there is a decrease in nuclear XBP-1s after tunicamycin treatment as well as a reduction in inositol-requiring enzyme 1α (IRE1α) and the activating transcription factor 6α (ATF6α) pathway, which are involved in UPR activation [77]. In addition, *Pik3r1*^−/−^ fibroblasts show blunted IRE1α-dependent XBP-1 splicing under ER stress. These results indicate that p85α is highly involved in resolving ER stress. When investigating the role of p85α in the UPR in the liver, L-*Pik3r1*^−/−^ mice have reduced IRE1α and ATF6α pathway activation after tunicamycin treatment in the liver, suggesting that the deletion of *Pik3r1* leads to impaired UPR [77]. Notably, upregulation of BRD7 has been shown to interact with XBP-1s and increase nuclear translocation of XBP-1s, which only occurs when p85α is present [71]. Immunoprecipitation experiments showed that BRD7 enhances p85α binding to XBP-1s [71]. In vivo studies showed that BRD7 protein levels are reduced in the liver of obese mice. However, BRD7 overexpression in the liver of obese mice restores XBP-1s nuclear translocation, which leads to a reduction in blood glucose levels in both obese and diabetic mouse models. This effect is attributed to the suppression of gluconeogenic gene transcription by XBP-1s [71,79]. Interestingly, BRD7 also interacts with p85β [70,71]. Overexpression of BRD7 in the liver of HFD-fed *Pik3r1*^−/−^ and *Pik3r2*^−/−^ mice improves glucose homeostasis. In *Pik3r2*^−/−^ mice, BRD7 overexpression increases insulin-stimulated IR and AKT phosphorylation, but it does not affect basal phosphorylation levels of AKT [72]. In contrast, in *Pik3r1*^−/−^ mice, BRD7 overexpression does not affect insulin-induced phosphorylation levels of IR and AKT. However, BRD7 overexpression leads to increased basal phosphorylation levels of AKT [72]. Thus, the role of BRD7-p85α and BRD7-p85β is distinct in the regulation of insulin sensitivity.

p85α has also been shown to directly interact with estrogen receptor (ERα) [80]. 17β-estradiol (E2) treatment increases ERα-associated PI3K activity in endothelial cells, leading to the activation of AKT and endothelial nitric oxide synthase (eNOS) [80]. The recruitment and activation of PI3K activity is independent of the transcriptional activity of ERα. Mice treated with E2 exhibited increased eNOS activity in endothelial cells and reduced vascular leukocyte accumulation after ischemia and reperfusion injury [4,80]. These phenotypes, however, were lost after treatment with PI3K inhibitors [80]. Thus, ERα-p85α interaction is a non-genomic action of ERα that plays a role in its cardioprotective function. Notably, in cortical neurons, E2 treatment also increases PI3K activity [80,81]. 

**Figure 4 ijms-24-12665-f004:**
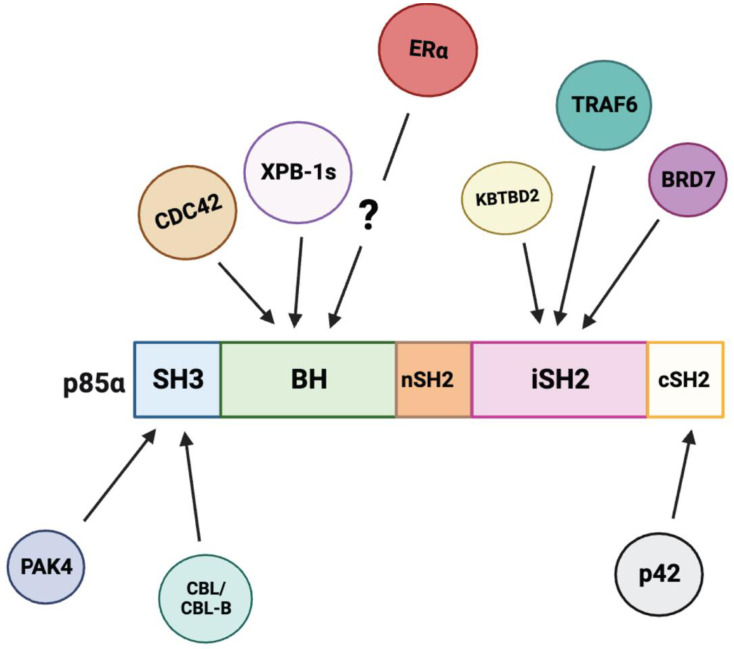
p85α interacts with various proteins through distinct domains. The biological effects of these interactions are listed in Table 2. ERα has been shown to bind to p85α without the presence of the SH3 or SH2 domains. However, the exact domain of p85α that associates ERα is unclear [80].

**Table 2 ijms-24-12665-t002:** Summary of PIK3R1-interacting proteins and their effects.

PIK3R1-Interacting Proteins	Location of Interaction on p85α	Effect of p85α Interaction	References
BRD7	iSH2 domain	BRD7 interaction leads to nuclear translocation of p85α [70]. Can reduce p85α levels in the cytosol and lead to reduced PI3K downstream signaling [70,71]. Enhances p85α binding to XBP-1s [72].	Chiu et al. [70] Park et al. [71]
PAK4	SH3 domain	PAK4 interaction relieves p110 catalytic subunit of PI3K. Reducing PAK4 levels in cells results in reduced phosphorylation of AKT [73].	King et al. [73]
CBL, CBL-B	SH3 domain	CBL and CBL-B interaction leads to p85α ubiquitination [65,66].	Bulut et al. [65] Fang et al. [66]
p42 (short isoform of EBP1)	cSH2 domain	p42 interaction leads to p85α degradation by HSP70/CHIP complex [67]. Inhibits PI3K activity [67].	Ko et al. [67]
TRAF6	iSH2 domain	TRAF6 polyubiquitinates p85α in its iSH2 domain, leading to formation of the TβR1 and p85α complex that activates PI3K and AKT [68].	Hamidi et al. [68]
KBTBD2	iSH2 domain	KBTBD2 interaction with p85α and CUL3 leads to p85α degradation [69]. KBTBD2 deletion in adipocytes leads to monomeric p85α accumulation which disrupts insulin signaling [69].	Zhang et al. [69]
CDC42	BH domain	CDC42 interaction is involved in cell migration through increasing filopodia formation, N-WASP-mediated decrease in actin stress fibers, and lowering focal adhesion complexes [76].	Jiménez et al. [76]
XBP-1s	BH domain	p85α interaction with XBP-1s leads to increased XBP-1s stability and nuclear translocation [53,77]. Depletion of p85α leads to impaired resolution of UPR [77].	Park et al. [53] Winnay et al. [77]
ERα	Unidentified	ERα interaction with p85α activates AKT and eNOS, which provides a cardioprotective effect.	Simoncini et al. [80] Hirsch et al. [4]

## 6. The Role of PIK3R1 in Hormonal Regulation of Metabolic Homeostasis 

The expression of *Pik3r1* is regulated by various stimuli, such as glucocorticoids and growth hormone. Glucocorticoids induce the expression of *Pik3r1* in both skeletal muscle and WAT (Figure 5) [57,82]. The glucocorticoid response element of mouse *Pik3r1* was identified in C2C12 myotubes using chromatin immunoprecipitation (ChIP) and reporter gene assays [82]. Thus, *Pik3r1* is a direct glucocorticoid receptor (GR) target gene [82]. Glucocorticoids are stress hormones whose major metabolic function is to maintain circulating glucose levels during stress, such as fasting. Glucocorticoids are also frequently used to treat various inflammatory and autoimmune diseases, such as arthritis, asthma, inflammatory bowel disease and lupus. Long-term glucocorticoid treatment, however, induces metabolic disorders, such as insulin resistance, hyperglycemia, muscle atrophy, hepatic steatosis, and dyslipidemia. In skeletal muscle, glucocorticoids induce protein degradation and prevent protein synthesis to utilize amino acids as gluconeogenic precursors [83]. Treating C2C12 myotubes with Dexamethasone (Dex, a synthetic glucocorticoid) reduces cell diameters. Overexpression of Pik3r1 in C2C12 myotube diameters mimics this glucocorticoid effect. In contrast, knockdown of Pik3r1 in C2C12 myotubes compromises Dex-decreased myotube diameters and protein synthesis [82]. Dex-induced expression of atrogenes in myotubes, such as *FoxO1*, *FoxO3* and *MuRF-1* are also reduced [82]. Reducing *Pik3r1* expression also significantly compromises the ability of Dex to inhibit AKT and p70S6 kinase activity and reduces glucocorticoid induction of IRS-1 phosphorylation at Ser307, which is associated with insulin resistance [82]. The role of striated muscle PIK3R1 in glucocorticoid responses was studied using M-*Pik3r1*^−/−^ mice. Dex treatment attenuates insulin-activated AKT activity in liver, epididymal white adipose tissue and gastrocnemius muscle in mice, and causes glucose and insulin intolerance in wild-type mice. Dex treatment also reduces protein synthesis in gastrocnemius muscle and causes skeletal muscle atrophy. These effects of Dex are all compromised in M-*Pik3r1*^−/−^ mice [54]. The ability of Dex to inhibit eIF2α phosphorylation and insulin-induced 4E-BP1 phosphorylation is also reduced in M-*Pik3r1*^−/−^ mice [82]. These in vivo results are mostly in agreement with the findings in C2C12 myotubes and highlight the role of PIK3R1 in mediating glucocorticoid-regulated glucose and protein homeostasis in striated muscle.

In WAT, glucocorticoids promote lipolysis, which generates glycerol and fatty acids. The former is the substrate of gluconeogenesis, whereas the latter is needed as energy sources during fasting. Dex treatment increases the levels of plasma free fatty acids (FFA), and ex vivo lipolysis assays found significant induction of glycerol release in WAT isolated from Dex-treated mice. These effects of Dex are reduced in adipose-specific knockout of *Pik3r1* mice (A-*Pik3r1*^−/−^), which were generated by crossing *Pik3r1^flox/flox^* mice with mice expressing adiponectin-Cre driver (Table 1) [57]. Dex treatment increases the phosphorylation of hormone sensitive lipase (Hsl) and perilipin 1 (Plin1) in WAT [57]. However, in A-*Pik3r1*^−/−^ mice, Dex treatment only elevates Hsl phosphorylation but not Plin1 phosphorylation. In the lipid droplets of WAT of A-*Pik3r1*^−/−^ mice, the levels of Dex-induced phospho-Hsl, phospho-Plin1, and catalytic and regulatory subunits of PKA are reduced [57]. This suggests that PIK3R1 plays a role, which is independent of its role in the regulation of PI3K, in Dex-increased PKA signaling in lipid droplets. The induction of WAT lipolysis by glucocorticoids is one of the mechanisms by which long-term glucocorticoid treatment induces hepatic steatosis and hypertriglyceridemia. In agreement with reduced WAT lipolysis in A-*Pik3r1*^−/−^ mice, these mice have improved Dex-induced hepatic steatosis and hypertriglyceridemia [57].

**Figure 5 ijms-24-12665-f005:**
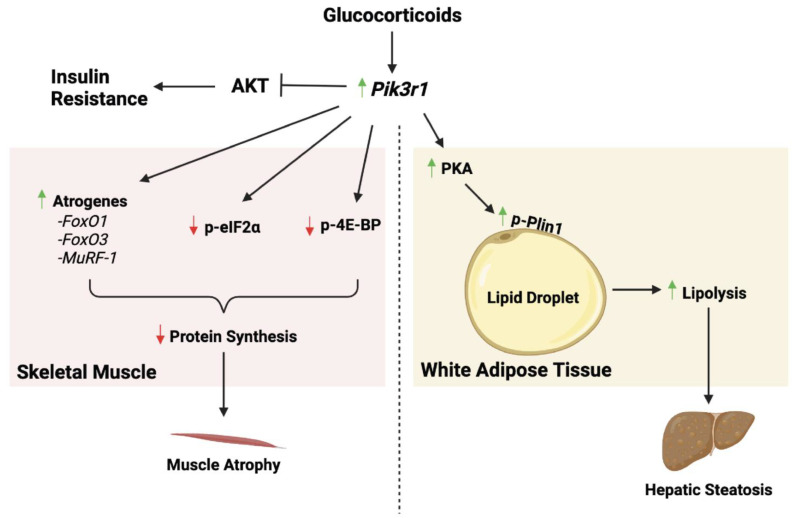
The role of Pik3r1 in glucocorticoid-regulated metabolism and insulin sensitivity in skeletal muscle and WAT. In skeletal muscle, glucocorticoids increase Pik3r1 expression, which results in an increased expression of atrogenes as well as a decrease in phosphorylated eIF2⍺ and phosphorylated 4E-BP. These lead to muscle atrophy. Glucocorticoids also induce Pik3r1 expression in WAT. In WAT, Pik3r1 is required for glucocorticoid-induced increase in PKA levels as well as phosphorylation of Plin1 in the lipid droplet. This process is involved in glucocorticoid-promoted lipolysis. The increased mobilization of fatty acids from lipolysis causes hepatic steatosis. Green up arrows indicate the induction and red down arrows indicate the reduction.

The expression of PIK3R1 is also induced by growth hormone (GH) in human vastus lateralis muscle and sub-cutaneous adipose tissue and mouse skeletal muscle and WAT [84,85,86,87]. Mice expressing bovine growth hormone (bGH mice) show increased *Pik3r1* mRNA and increased p85α protein levels in WAT [84]. On the other hand, mice that are chronically deficient in growth hormone (*lit/lit* mice) have lower p85α protein expression in WAT along with improved insulin sensitivity [84]. It is suggested that the improved insulin sensitivity seen in *lit/lit* mice is due to decreased p85α levels as there is strong correlation between *Pik3r1* and *insulin growth-like factor-1* (*Igf-1*), a target gene of GH, mRNA levels [84]. Under insulin treatment, *lit/lit* mice have a 207% increase in IRS-1-mediated PI3K activity, illustrating that GH regulates PI3K activation through insulin and IRS-1 in WAT [84]. Separately, another study determined that mice with liver-specific deletion of *Igf-1* show GH-induced insulin resistance contributed by excess p85α produced in the skeletal muscle [88]. Treatment with a growth hormone-releasing hormone antagonist in liver-specific deletion of *Igf-1* mice reverses the increased p85α production in skeletal muscle and insulin resistance phenotype [88]. GH injection results in an increased p85α expression and insulin resistance in wild-type and *Pik3r2*^−/−^ mice. Compared to wild-type and *Pik3r2*^−/−^ mice, insulin-stimulated PI3K activity is somewhat greater in the *Pik3r1*^+/−^ mice. Moreover, the administration of GH blunted insulin-stimulated IRS-1-associated PI3K activity in the wild-type and *Pik3r2*^−/−^ mice but failed to affect PI3K activity in the *Pik3r1*^+/−^ mice [88]. Overall, growth hormone regulates p85α expression to affect insulin sensitivity in WAT and skeletal muscle. 

## 7. Conclusions and Perspectives 

PI3K regulates many facets of metabolic homeostasis. As a regulatory subunit of PI3K, PIK3R1 plays an important role in the modulation of PI3K activity. Many transgenic- and tissue-specific *Pik3r1* knockout animals have been created to study the in vivo function of *Pik3r1*. Various PI3K-independent functions of PIK3R1 that are involved in metabolic regulation are also identified. Despite these progresses, many aspects of the role of PIK3R1 in the regulation of metabolic homeostasis remain to be explored. For example, how post-translational modifications of PIK3R1 and PIK3R1-interacting proteins affect whole-body metabolism remains to be evaluated. Many studies of PIK3R1 function focus on its role in tumorigenesis and regulation of cancer cell physiology. While we did not discuss those aspects in this article, it is plausible to foresee that some of these mechanisms are employed to regulate metabolic homeostasis. Overall, we expect that more mechanisms regulating PIK3R1 activity in metabolic homeostasis and the PI3K-independent PIK3R1 metabolic functions will be identified in future studies.

## Figures and Tables

**Figure 2 ijms-24-12665-f002:**
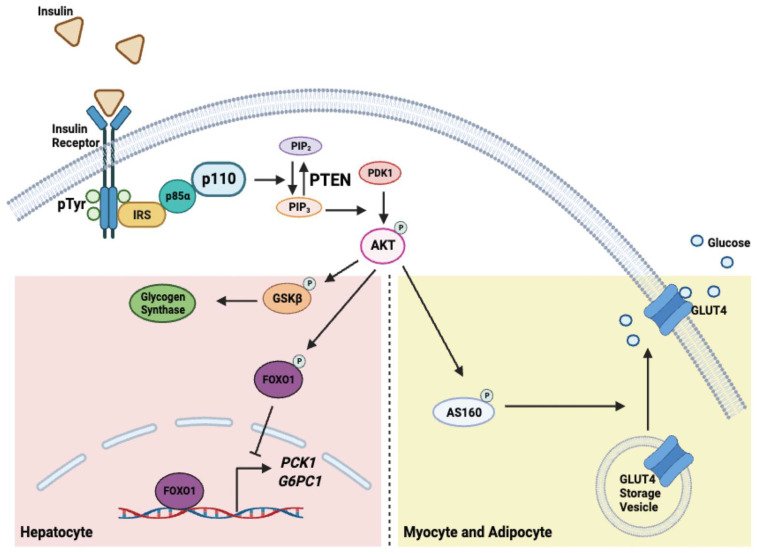
The role of p85⍺ in insulin signaling to modulate glucose homeostasis in hepatocytes, myocytes, and adipocytes. Activated insulin receptor leads to autophosphorylation of its own tyrosine residues that allow the binding of IRS. Tyrosine phosphorylation of IRS by IR then recruits p85⍺. This releases the inhibitory effect of p85⍺ on p110, allowing p110 to convert PIP_2_ to PIP_3_. PIP_3_ recruits PDK1 and AKT. PDK1 phosphorylates and activates AKT. In hepatocytes, activated AKT inhibits GSKβ and activates glycogen synthase. In addition, activated AKT phosphorylates FOXO1, sequestering it in the cytosol and preventing it from binding to the promoter to activate transcription of gluconeogenic genes *PCK1* and *G6PC1*. In myocytes and adipocytes, activated AKT phosphorylates AS160, which promotes the translocation of GLUT4 to the plasma membrane.

**Figure 3 ijms-24-12665-f003:**
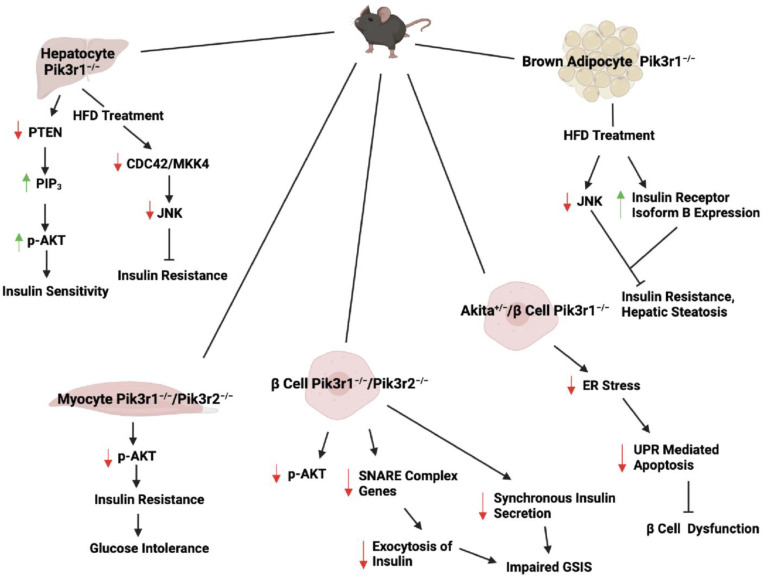
Cell type- and tissue-specific *Pik3r1* knockout mice reveal its metabolic functions. Liver-specific Pik3r1 knockout mice have improved hepatic and whole-body insulin sensitivity due to increased hepatic AKT activity. On HFD treatment, these mice are protected against c-Jun N-terminal kinase (JNK)-mediated insulin resistance. In myocyte-specific *Pik3r1* knockout mice, AKT activity is not affected. However, M-*Pik3r1*^−/−^/*Pik3r2*^−/−^ mice have impaired AKT activity, resulting in insulin resistance and glucose intolerance. β *cell Pik3r1*^−/−^/*Pik3r2*^−/−^ mice have impaired glucose-stimulated insulin secretion (GSIS) due to reduced exocytosis of insulin and induced synchronous insulin secretion. Akita^+/−^/β cell *Pik3r1*^−/−^ mice have reduced β cell dysfunction due to decrease in ER-mediated apoptosis. Brown adipocyte-specific *Pik3r1* knockout mice have attenuated HFD-induced insulin resistance and hepatic steatosis due to reduced JNK activation and enhanced insulin receptor isoform B expression. Green up arrows indicate the induction and red down arrows indicate the reduction.

## Data Availability

Not applicable.

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
