# Peer review of "The Role of PIK3R1 in Metabolic Function and Insulin Sensitivity"

_ijms, 2023, doi:10.3390/ijms241612665_

Round 1

Reviewer 1 Report

Although the manuscript provides a well-written overview of the field, it lacks significant effort from the authors. With only one figure throughout the entire manuscript, it fails to enhance readability and accessibility for the target audience. To improve the quality of the manuscript, the authors should focus on incorporating more figures.

Here are specific sections that require figures:

·      Insulin Signaling: Given the numerous proteins mentioned in the text, such as IRS, AKT, PDK, FOXO, and mTOR, it is essential to create a corresponding diagram for this section.

·      Metabolic Functions of PIK3R1 in vivo: Please include a figure dedicated to illustrating the content in this section.

·      PIK3R1-interacting proteins and PI3K independent function of PIK3R1: This section would benefit from a figure that visually represents the information discussed.

·      The Role of PIK3R1 in Hormonal Regulation of Metabolic Homeostasis: A figure should be created to complement the content covered in this section.

By addressing these recommendations and incorporating additional figures, the manuscript will become more comprehensive, visually appealing, and better suited to the needs of the readership.

Reviewer 2 Report

Dear Editors,

The study shows an extensive review about the relevance of PI3KR1 in the metabolic functions to maintenance of body homeostasis. They discuss diverse studies involving whole body knockout (KO) models or tissue-specific KO animals to elucidate the real role of this protein in the cellular metabolism of muscle, liver, adipose tissue, B cell pancreatic, as well as homeostasis in general. Additionally, they discuss some aspects related to the mutation of this protein and its effects on metabolic homeostasis, as well as its influence on some hormonal effects, such as glucocorticoids and GH.

Some minor points should be improved, as follow:

1- Introduction line 3 – correct the word phosphAtidylinositol.

2- Introduction line 3 – correct the word phosphoinositides.

3- Introduction line 4 – the authors have used the words “several reviews”, but they have cited only 2 references.

4- Introduction line 55 – the verbal conjugation is wrong in the sentence after [7] reference.

5- Introduction line 78 – the authors have used the abbreviation SUMO without a previous description of the meaning.

6- Line 202 – remove the verb “to be” after the word these.

In general, I believe the English language is readable and well-written. There are a few typo errors.

Reviewer 3 Report

The authors provide a well-written and comprehensive review on the role of the PIK3R1 protein in whole-body metabolic regulation. The manuscript focuses on PI3Ks and their major role downstream the insulin receptor: in particular, the authors highlight the importance of the regulatory subunit PIK3R in insulin signaling pathway and the impact that mutations in the PIK3R1 gene have in humans and mice.

Despite the broad existing literature on PI3Ks and their regulatory proteins, the manuscript is relevant for researchers in the field as it focuses on PIK3R1 protein and provide a thorough analysis of its role in metabolic regulation.

However, I would recommend the authors to address the following minor points to improve the quality of their manuscript:

1)      Insulin signaling pathway is complex and numerous proteins are involved downstream the insulin receptor (IR). To help the reader, I suggest that the authors add a new figure with a schematic of PIK3R1’s modulation of IRS and other proteins downstream IR (listed in paragraph nr 2).

2)      Since the review analyses the impact of gene mutation in human and murine metabolic regulation, the authors should include the location of gene PIK3R2 (in humans and mice).

3)      Within the paragraph on PIK3R1 structure, the authors could refer to “Figure 1” only once at the beginning (there are redundancies at lines 52-83).

4)      A short sentence in paragraph nr 3 may be added at line nr 141 to briefly introduce conditional mice studies that are listed in paragraphs 3.1-3.4.

5)      I would suggest that the authors add (and possibly discuss) the following relevant publications on the topic:

o   a comprehensive review on PI3Ks and their role in metabolism (PMID: 17641274);

o   additional literature on the oncogenic activity induced by loss of PIK3R1 (PMID: 28630349 and 30961830).

Reviewer 4 Report

To the authors:

The review reports the role of the PI3K regulatory subunit p85a in metabolism and insulin sensitivity by mainly reporting the characteristics of different KO animal models. In addition, the proteins interacting with p85are described. The review is easy to read and the topic well treated. 

- In addition to the description the characteristics of the different KO models, a clear conclusion on p85a role in the different metabolic tissues (liver, adipose tissue, muscle, b cells in pancreas) will be more informative. 

- The role of p85a in metabolic functions is driven by the results obtained in tissues with selective deletion of the regulatory subunit. It will also be of interest to discuss whether pharmacological tools towards p85a are under investigation and their relevance in metabolic diseases. For example, will selective PI3kinase inhibitors of therapeutical interest for metabolic diseases? 

- The title could be ameliorated, “metabolic regulation” is not so clear.  Metabolic functions would be more appropriate. 

Round 2

Reviewer 1 Report

Authors have done all suggested reviews, I recommend the article for publication.